# EFFECTIVE ABSTRACT REASONING WITH DUAL-CONTRAST NETWORK

**Tao Zhuo, Mohan Kankanhalli**
School of Computing, National University of Singapore
`zhuotao@nus.edu.sg, mohan@comp.nus.edu.sg`

## ABSTRACT

As a step towards improving the abstract reasoning capability of machines, we aim to solve Raven's Progressive Matrices (RPM) with neural networks, since solving RPM puzzles is highly correlated with human intelligence. Unlike previous methods that use auxiliary annotations or assume hidden rules to produce appropriate feature representation, we only use the ground truth answer of each question for model learning, aiming for an intelligent agent to have a strong learning capability with a small amount of supervision. Based on the RPM problem formulation, the correct answer filled into the missing entry of the third row/column has to best satisfy the same rules shared between the first two rows/columns. Thus we design a simple yet effective Dual-Contrast Network (DCNet) to exploit the inherent structure of RPM puzzles. Specifically, a rule contrast module is designed to compare the latent rules between the filled row/column and the first two rows/columns; a choice contrast module is designed to increase the relative differences between candidate choices. Experimental results on the RAVEN and PGM datasets show that DCNet outperforms the state-of-the-art methods by a large margin of 5.77%. Further experiments on few training samples and model generalization also show the effectiveness of DCNet. Code is available at https://github.com/visiontao/dcnet.

## 1 INTRODUCTION

Abstract reasoning capability is a critical component of human intelligence, which relates to the ability of understanding and interpreting patterns, and further solving problems. Recently, as a step towards improving the abstract reasoning ability of machines, many methods (Santoro et al., 2018; Zhang et al., 2019a;b; Zheng et al., 2019; Zhuo & Kankanhalli, 2020) are developed to solve Raven's Progress Matrices (RPM) (Domino & Domino, 2006; Raven & Court, 1938), since it is widely believed that RPM lies at the heart of human intelligence. As the example shown in Figure 1, given a $3 \times 3$ problem matrix with a final missing piece, the test taker has to find the logical rules shared between the first two rows or columns, and then pick the correct answer from 8 candidate choices to best complete the matrix. Since the logical rules hidden in RPM questions are complex and unknown, solving RPM with machines remains a challenging task.

As described in (Carpenter et al., 1990), the logical rules applied in a RPM question are manifested as visual structures. For a single image in the question, the logical rules could consist of several basic attributes, *e.g.,* shape, color, size, number, and position. For the images in a row or column, the logical rules could be applied row-wise or column-wise and formulated with an unknown relationship, *e.g.,* AND, OR, XOR, and so on (Santoro et al., 2018; Zhang et al., 2019a). If we can extract the explicit rules of each question, the problem can be easily solved by using a heuristics-based search method (Zhang et al., 2019a). However, given an arbitrary RPM question, the logical rules are unknown. What's worse - even the number of rules is unknown. As a result, an intelligent machine needs to simultaneously learn the representation of these hidden rules and find the correct answer to satisfy all of the applied rules.

With the success of deep learning in computer vision, solving RPM puzzles with neural networks has become popular. Because the learned features might be inconsistent with the logical rules, many supervised learning methods, *e.g.,* DRT (Zhang et al., 2019a), WReN (Santoro et al., 2018) and

LEN (Zheng et al., 2019), MXGNet (Wang et al., 2020) and ACL (Kim et al., 2020), not only use the ground truth answer of each RPM question but also the auxiliary annotations (such as logical rules with shape, size, color, number, AND, OR, XOR) to learn the appropriate feature representation. Although auxiliary annotations provide lots of priors about how the logical rules are applied, noticeable performance improvement is not always obtained on different RPM problems, such as in the results reported in Table 1. Moreover, such a learning strategy requires additional supervision. When auxiliary annotations are not available, it will fail to boost the performance. For example, DRT (Zhang et al., 2019a) cannot be applied to PGM dataset (Santoro et al., 2018) for the lack of structure annotations. To overcome the constraint of using auxiliary annotations, a recent method CoPINet (Zhang et al., 2019b) only uses the ground truth answer of each question. Meanwhile, to produce the feature representation of hidden rules, CoPINet assumes there are at most $N$ attributes in each problem, and each of which is subject to the governance of $M$ rules. However, due to $N$ and $M$ being unknown for arbitrary RPM problems, such an assumption is still too strong.

In this work, we aim to learn the abstract reasoning model by using only the ground truth answer of each question, and there is not any assumption about the latent rules. According to the RPM problem formulation (Carpenter et al., 1990), it can be concluded that finding the correct answer of a RPM puzzle mainly depends on two contrasts of rules: (1) compare the hidden rules between the filled row/column and the first two rows/columns to check whether they are the same rules; (2) compare all candidate choices to check which one best satisfies the hidden rules. Unlike making specific assumptions about latent rules that are only valid for particular cases, the above two contrasts are general properties of all RPM problems.

Considering above two contrasts, we propose a simple yet effective Dual-Contrast Network (DC-Net) to solve RPM problems. Specifically, a rule contrast module is used to compute the difference between the filled row/column and the first two rows/columns, which checks the difference between latent rules. Additionally, the second choice contrast module is used to increase the relative differences of all candidate choices, which helps find the correct answer when confusingly similar choices exist. Experiments on two major benchmark datasets demonstrate the effectiveness of our method. In summary, our main contributions are as follows:

- We propose a new abstract reasoning model on RPM with only ground truth answers, *i.e.* there are not any assumptions or auxiliary annotations about the latent rules. Compared to previous methods, the problem setting of our method is more challenging, as we aim for an intelligent agent to learn a strong model with a small amount of supervision.

- We propose a simple yet effective Dual-Contrast Network (DCNet) that consists of a rule contrast module and a choice contrast module. By exploiting the inherent structures of each RPM with basic problem formulation, robust feature representation can be learned.

- Experimental results on RAVEN (Zhang et al., 2019a) and PGM (Santoro et al., 2018) datasets show that our DCNet significantly improves the average accuracy by a large margin of 5.77%. Moreover, from the perspective of few-shot learning, DCNet outperforms the state-of-the-art method CoPINet (Zhang et al., 2019b) by a noticeable margin when few training samples are provided, see Table 3 and 4. Further experiments on model generalization also show the effectiveness of our method, see Table 5 and 6.

## 2 RELATED WORK

### 2.1 VISUAL REASONING

One of the most popular visual reasoning tasks is Visual Question Answering (VQA) (Antol et al., 2015; Zellers et al., 2019; Fan et al., 2020). To aid in the diagnostic evaluation of VQA systems, Johnson et al. (2017) designed a CLEVR dataset by minimizing bias and providing rich ground-truth representations for both images and questions. It is expected that rich diagnostics could help better understand the visual reasoning capabilities of VQA systems. Recently, to understand the human actions in videos, Zhou et al. (2018) proposed a temporal relational reasoning network to learn and reason about temporal dependencies between video frames at multiple time scales. Besides, for explainable video action reasoning, Zhuo et al. (2019) proposed to explain performed actions by recognizing the semantic-level state changes from a spatio-temporal video graph with pre-defined rules. Different from these visual tasks, solving RPM puzzles depends on sophisticated logical rules

acquiring human intelligence (Raven & Court, 1938; Santoro et al., 2018; Zhang et al., 2019a). Therefore, it is expected that solving RPM problems with machines could help better understand and perhaps improve the abstract reasoning ability of contemporary computer vision systems.

## 2.2 COMPUTATIONAL MODELS ON RPM

The early methods on RPM (McGreggor & Goel, 2014; McGreggor et al., 2014; Mekik et al., 2018) often compute the feature similarity of images with a hand-crafted representation. Besides, structural affinity (Shegheva & Goel, 2018) with graphical models are also used in RPM problems. Due to the lack of large-scale dataset, the early methods are only evaluated on a small dataset. Recently, with the introduction of deep learning in pattern recognition (Deng et al., 2009), solving RPM with deep neural networks has becomes the main approach. For automatic RPM generation, Wang & Su (2015) introduced an abstract representation method for RPM by using the first-order logic formulae, and they applied three categories of relations (*i.e.* unary, binary, and ternary). Hoshen & Werman (2017) first trained a CNN model to measure the IQ of neural networks in a simplistic evaluation environment. Santoro et al. (2018) measured the abstract reasoning capability of machines with several popular neural networks, such as LSTM, ResNet and Wild-ResNet. Besides, Zhuo & Kankanhalli (2020) studied the effect of ImageNet pre-training (Deng et al., 2009) for RPM problems and proposed a novel unsupervised abstract reasoning method by designing a pseudo target.

To further improve the feature representation with auxiliary annotations, Santoro et al. (2018) proposed a WReN network that formulates pair-wise relations between the problem matrix and each individual choice in embedding space, independent of the other choices. Zhang et al. (2019a) generated a RAVEN dataset with the structure annotations for each RPM problem and proposed a Dynamic Residual Tree (DRT) method. In addition, Zheng et al. (2019) proposed a Logic Embedding Network (LEN) with distracting features, which also uses the auxiliary annotations to boost the performance and designed a teacher model to control the learning trajectories. MXGNet (Wang et al., 2020) is a multi-layer graph neural network for multi-panel diagrammatic reasoning tasks. For better performance, MXGNet also uses auxiliary annotations for model training. ACL (Kim et al., 2020) uses symbolic labels to generate new RPMs with the same problem semantics but with different attributes (*e.g.,* shapes and colors). Discarding the use of auxiliary annotations, CoPINet (Zhang et al., 2019b) only utilizes the ground truth answer of each RPM question. By combining contrasting, perceptual inference, and permutation invariance, CoPINet achieves the state-of-the-art performance. However, it assumes the maximum number of attributes and rules is known. Different from the "objective-level contrast" in (Zhang et al., 2019b) that treats the complete problem matrix (9 images, including a filling choice) as an object for comparison, our model consists of a rule contrast module and a choice contrast module and we only compare the features of different rows/columns (3 images). Thus our method is more effective to exploit the inherent structure of RPM on rows/columns. Moreover, no assumptions about attributes and number of rules are made.

## 3 DUAL-CONTRAST NETWORK

A typical RPM problem usually consists of 16 images, including a $3 \times 3$ problem matrix with a final missing piece and 8 candidate answer choices. Solving a RPM problem is to find the correct answer from 8 candidate choices to best complete the problem matrix. Given $N$ RPM questions $\mathcal{X} = \{(\boldsymbol{x}_1, y_1), \cdots, (\boldsymbol{x}_N, y_N)\}$, where in each question $(\boldsymbol{x}_i, y_i)$ $(i \in [1, N])$, $\boldsymbol{x}_i$ denotes 16 images and $y_i$ represents the ground truth answer, we aim at learning an abstract reasoning model on $\mathcal{X}$ that finds the correct answer. Different from previous methods (Santoro et al., 2018; Zheng et al., 2019) that use auxiliary annotations on hidden rules, we only use the ground truth answer of each question. Because of less information use for supervision, the problem setting of our method is more challenging.

Based on the RPM problem formulation (Carpenter et al., 1990), given a RPM problem with a set of rules applied either row-wise or column-wise, the correct answer filled into the third row/column has to best satisfy the same rules shared between the first two rows/columns. Thus, it follows that the correct answer of a RPM question has to satisfy two contrasts of hidden rules: (1) the third row/column filled with a choice whether shares the same rules as that of the first two rows/columns; (2) the choice is the best one to complete the matrix when compared to the other candidate choices. Inspired by these two inherent contrasts of RPM problem formulation, we propose a dual-contrast

model for abstract reasoning. Specifically, a rule contrast module is designed to compare the hidden rules between filled row/column and the first two rows/columns; a choice contrast module is designed to increase the relative differences of candidate choices.

### 3.1 RULE CONTRAST MODULE

As described in (Carpenter et al., 1990), the logical rules in a RPM problem could be applied either row-wise or column-wise. However, for an arbitrary RPM problem, which logical rules have been applied is unknown. Thus, we equally consider the row-wise and column-wise features to represent the hidden rules and we process them in the same way. In order to simplify the description, we only take the row-wise feature as example in next.

Given a RPM problem $\boldsymbol{x}_i$, we iteratively fill each candidate choice into the missing piece of the problem matrix, forming a new problem of 10 rows/columns. Specifically, $\boldsymbol{r}_{i,j}$ denotes the $j$-th row of $\boldsymbol{r}_i$, $j \in \{1, \cdots, 10\}$, $\boldsymbol{x}_{i,j+6}$ represents $(j+6)$-th image of $\boldsymbol{x}_i$. If the row $\boldsymbol{r}_{i,j^*}$ share the same latent rules with $\boldsymbol{r}_{i,1}$ and $\boldsymbol{r}_{i,2}$, it will be regarded as the correct answer. Thus, by comparing the features among the last eight rows and the first two rows, the correct answer $j^*$ can be found.

Suppose $f(\cdot)$ is the embedding feature representation of the latent rules of a row, and then the row-wise feature of $\boldsymbol{r}_{i,j}$ is denoted as $f(\boldsymbol{r}_{i,j})$. Due to the logical rules applied in different RPM problems are often different, which would lead to inaccurate feature representation. To alleviate this issue, we explicitly check how the latent rules are satisfied for each row. Based on the fact that $f(\boldsymbol{r}_{i,1})$, $f(\boldsymbol{r}_{i,2})$, and $f(\boldsymbol{r}_{i,j^*})$ share the same logical rules, they can be clustered into the same group while the features of other rows in another group. Therefore, by subtracting the clustering centroid of $f(\boldsymbol{r}_{i,1})$ and $f(\boldsymbol{r}_{i,2})$, the rule contrast feature $g(\boldsymbol{r}_{i,j})$ on $j$-th row can be extracted to compare the relative rule difference of different rows as:

$$g(\boldsymbol{r}_{i,j}) = f(\boldsymbol{r}_{i,j}) - 0.5 * (f(\boldsymbol{r}_{i,1}) + f(\boldsymbol{r}_{i,2})) \tag{1}$$

where $j \geq 3$. Since the proposed rule contrast module explicitly compares the latent rules of each row, which matches the inherent structure of RPM problems. Hence, our model can adapt to the various logical rules, helping improve the model generalization.

### 3.2 CHOICE CONTRAST MODULE

The correct answer in a RPM problem satisfies not only the same rules shared between the first two rows but also better fits the third row when compared to all other candidate choices. Given a set of candidate choices for a RPM question, the designed model has to distinguish which one is the correct answer. However, to make RPM questions difficult, there are often some confusing choices that are similar to the correct answer (Zhang et al., 2019a; Santoro et al., 2018). Moreover, the confusing similarities can be different for different problems. Therefore, directly learning the model without comparing all of the candidate choices would lead to poor performance.

In order to distinguish the correct answer from similar candidate choices, we use a choice contrast module to increase the relative differences among the candidate rows (*i.e.* filling the third row with a candidate choice). By reducing the centroid of all candidate choices in a RPM problem, the choice contrast module is formulated as:

$$h(\boldsymbol{r}_{i,j}) = g(\boldsymbol{r}_{i,j}) - \phi(\frac{1}{8} \sum_{j=3}^{10} g(\boldsymbol{r}_{i,j})) \tag{2}$$

where $h(\boldsymbol{r}_{i,j})$ denotes the relative feature of the third row with a candidate choice, $j \geq 3$ (corresponding to eight candidate rows), $\phi(\cdot)$ is an adaptive feature block that consists of a CNN layer and a batch normalization layer (Ioffe & Szegedy, 2015).

Notice that the latent rules have be checked with the rule contrast, the choice contrast is only applied on the last eight rows (*i.e.* removing the first two rows). Based on such an explicit choice contrast module, the proposed model is able to handle various confusing similarities of candidate choices in different RPM problems.

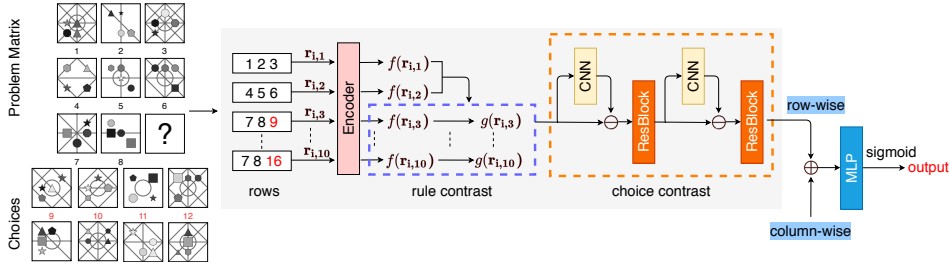

Figure 1: The architecture of our Dual-Contrast Network (DCNet). The number in the boxes denotes the corresponding image and the encoder module consists of a CNN layer (with ReLU and batch normalization (Ioffe & Szegedy, 2015)), a maxpooling layer and a residual block (He et al., 2016). Besides, the two streams of DCNet share the same parameters and they are used to extract the row-wise and column-wise features, respectively. Then these two features are added and further fed into a two-layer MLP module for the answer prediction. The correct answer for this problem is 13.

## 3.3 INFERENCE

Since the logical rules applied in row-wise or column-wise are unknown, we equally consider both conditions. After obtaining the dual-contrast features of the rows and columns, the final answer can be predicted with a classification module. By directly feeding the sum of the row-wise feature $h(\boldsymbol{r}_{i,j})$ and the column-wise feature $h(\boldsymbol{c}_{i,j})$ into a two-layer MLP (two fully-connected layers with a ReLU and a dropout layer (Srivastava et al., 2014)), the probability $\tilde{y}_{i,j}$ of each row/column is estimated as:

$$\tilde{y}_{i,j} = \psi(h(\boldsymbol{r}_{i,j}) + h(\boldsymbol{c}_{i,j})) \tag{3}$$

where $j \in \{3, \cdots, 10\}$ corresponds to 8 candidate choices, $\psi(\cdot)$ represents the projection function of using the MLP module. The choice with the highest score is selected as the correct answer.

## 3.4 TRAINING

The correct answer of a RPM question is obtained by comparing the probabilities of all candidate choices, and thus finding the correct answer can be considered as a ranking task. Hence, we set a label 1 to the correct answer while set the label 0 to the other choices. Besides, in order to further increase the difference between the correct answer and the other candidate choices, the estimated probability $\tilde{y}_{i,j}$ of each choice are concatenated and a sigmoid function $\delta(\cdot)$ is employed for normalization. In addition, a Binary Cross Entropy (BCE) loss function $\mathcal{L}$ is employed for the model training. Given a dataset $\mathcal{X}$ that consists of $N$ questions, the training loss $\mathcal{L}$ is computed as:

$$\mathcal{L} = \sum_{i=1}^{N} \sum_{j=1}^{8} -[y_{i,j} \log(\delta(\tilde{y}_{i,j})) + (1 - y_{i,j}) \log(1 - \delta(\tilde{y}_{i,j}))], \tag{4}$$

where the index $j \in \{1, \cdots, 8\}$ starts from 1 for the sake of clarity.

## 3.5 NETWORK ARCHITECTURE

Based on the rule contrast and the choice contrast modules, we propose a Dual-Contrast Network (DCNet) for abstract reasoning on RPM. The overview of the proposed DCNet architecture is depicted in Figure 1, which consists of two streams and an MLP fusion layer. As the row-wise and column-wise features are equally considered for the final answer prediction, these two streams share the same parameters. Given a RPM problem, by iteratively filling each choice to the missing piece, 10 rows and 10 columns are formed, respectively. Then an encoder of DCNet is used to extract the row-wise and column-wise features, representing latent rules of each row/column. This is followed by a dual-contrast module to compare the latent rules and increase the relative differences of candidate choices. Finally, a two-layer MLP module is used for the correct answer prediction.

## 4 EXPERIMENTS

### 4.1 IMPLEMENTATION DETAILS

Similar to the previous works (Zhang et al., 2019b; Zheng et al., 2019; Wang et al., 2020), we conduct experiments on the RAVEN (Zhang et al., 2019a) and PGM (Santoro et al., 2018). All images in our experiments are resized to a fixed size of $96 \times 96$. During the training phase, a mini-batch size of 32 with Adam optimizer (Kingma & Ba, 2015) is employed to learn the network parameters, and the learning rate is set to 0.001 and fixed. In all experiments, the network architecture and training parameters are kept the same to demonstrate the effectiveness of DCNet. In addition, all models are trained and evaluated on a single GPU of NVIDIA GeForce 1080 Ti with 11 GB memory.

### 4.2 DATASETS

**RAVEN dataset** (Zhang et al., 2019a) consists of 70K RPM problems, equally distributed in 7 distinct figure configurations: *Center*, *2*2Grid*, *3*3Grid*, *Left-Right (L-R)*, *Up-Down (U-D)*, *Out-InCenter (O-IC)* and *Out-InGrid (O-IG)*. In each configuration, the dataset is randomly split into three parts, 6 folds for training, 2 for validation, and the remaining 2 for testing. In addition, there is an average of 6.29 rules for each problem, and those rules are applied only row-wise.

**PGM dataset** (Santoro et al., 2018) consists of 8 subdatasets. Compared to RAVEN dataset that consists of shapes only, PGM dataset is constructed with both lines and shapes. Besides, some lines or shapes might be manifested as background for hidden rules in PGM. In our experiment, we adopt the same experimental setup of CoPINet (Zhang et al., 2019b) and we mainly report the testing accuracy of the neutral regime in PGM, since it corresponds most closely to traditional supervised regimes. In total, the neural regime subset consists of 1.42M RPM problems, with 1.2M for training, 20K for validation and 200K for testing. There is an average of 1.37 rules for each problem, and those rules are applied either row-wise or column-wise.

### 4.3 BASELINE MODELS

We compare the proposed method with several available results of the state-of-the-art approaches, including LSTM (Shi et al., 2015), CNN (Hoshen & Werman, 2017), ResNet-50 (He et al., 2016), WReN (Santoro et al., 2018), Wild-ResNet (Santoro et al., 2018), ResNet-18+DRT (Zhang et al., 2019a), LEN (Zheng et al., 2019), CoPINet (Zhang et al., 2019b), MXGNet (Wang et al., 2020), and ACL (Kim et al., 2020). In Zheng et al. (2019), there are several models based on different training configurations. For the sake of fair comparison, we only report the performance of LEN with or without auxiliary annotations, due to the fact that T-LEN and teacher model in (Zheng et al., 2019) require external supervision (except using auxiliary annotations in training loss) for model training.

### 4.4 PERFORMANCE ANALYSIS

**Overall Performance.** In Table 1, it can be seen that DCNet significantly improves the average accuracy by a large margin of absolute 5.77% when compared to MXGNet, *i.e.* 81.08% *vs.* 75.31%. Moreover, without auxiliary annotations on RAVEN, DCNet achieves the best performance and it outperforms CoPINet by 2.16%, *i.e.* 93.58% *vs.* 91.42%. MLRN (Jahrens & Martinetz, 2020) is elaborately designed for PGM and it achieves an accuracy of 98.03%. However, MLRN performs poorly on RAVEN with the same network architecture and hyper-parameters. WReN, LEN and MXGNet greatly boost the performance with auxiliary annotations and they achieve better accuracy than other models, but they require additional supervision. Removing the auxiliary annotations for fair comparison, DCNet slightly outperforms LEN by 0.47%.

**Performance on RAVEN Dataset.** Table 2 shows the testing accuracy of each model on the 7 distinct figure configurations of RAVEN dataset. The performance of the popular CNN and LSTM models is quite poor on the RAVEN dataset, which is less than 40%. Although WReN obtains double accuracy with auxiliary annotations, but it is still poor (33.97%). For another method LEN, auxiliary annotations make the performance even worse, *i.e.* from 72.90% to 53.40%. Such a performance decrease is also observed in (Zhang et al., 2019a). In contrast, without any auxiliary annotations for

Table 1: Testing accuracy of different models on RAVEN and neutral regime of PGM. Aux means auxiliary annotations. Avg represents the average accuracy, DCNet-RC denotes removing the rule contrast module and DCNet-CC denotes removing choice contrast module in our method. "-" indicates the results are not reported in the published papers. "#" represents the result is obtained from Zhuo & Kankanhalli (2020). "*" represents the result is obtained by running the released code of Jahrens & Martinetz (2020) with its default parameters.

| Method | Aux | Avg | RAVEN | PGM |
|---|---|---|---|---|
| ResNet-18+DRT Zhang et al. (2019a) | ✓ | - | 59.56 | - |
| WReN+Aux Santoro et al. (2018) | ✓ | 55.44 | 33.97 | 76.90 |
| LEN+Aux Zheng et al. (2019) | ✓ | 70.85 | 59.40 | 82.30 |
| MXGNet+Aux Wang et al. (2020) | ✓ | - | - | **89.60** |
| ACL Kim et al. (2020) | ✓ | - | **93.71** | - |
| LSTM Zhang et al. (2019b) | | 24.44 | 13.07 | 35.80 |
| CNN Zhang et al. (2019b) | | 34.99 | 36.97 | 33.00 |
| WReN Santoro et al. (2018) | | 40.10 | 17.94 | 62.60 |
| Wild-ResNet Santoro et al. (2018) | | - | - | 48.00 |
| ResNet-50 Santoro et al. (2018) | | 64.13 | 86.26# | 42.00 |
| MLRN Jahrens & Martinetz (2020) | | 55.33 | 12.50* | **98.03** |
| LEN Zheng et al. (2019) | | 70.50 | 72.90 | 68.10 |
| CoPINet Zhang et al. (2019b) | | 73.90 | 91.42 | 56.37 |
| MXGNet Wang et al. (2020) | | 75.31 | 83.91 | 66.70 |
| DCNet-RC | | 78.10 | 92.74 | 63.45 |
| DCNet-CC | | 47.12 | 36.47 | 57.76 |
| DCNet | | **81.08** | **93.58** | 68.57 |

Table 2: Testing accuracy of different models on different figure configurations in RAVEN dataset.

| Method | Aux | Avg | Center | 2*2Grid | 3*3Grid | L-R | U-D | O-IC | O-IG |
|---|---|---|---|---|---|---|---|---|---|
| WReN+Aux Santoro et al. (2018) | ✓ | 33.97 | 58.38 | 38.89 | 37.70 | 21.58 | 19.74 | 38.84 | 22.57 |
| LEN+Aux Zheng et al. (2019) | ✓ | 59.40 | 71.10 | 45.90 | 40.10 | 63.90 | 62.70 | 67.30 | 65.20 |
| ResNet-18+DRT Zhang et al. (2019a) | ✓ | 59.56 | 58.08 | 46.53 | 50.40 | 65.82 | 67.11 | 69.09 | 60.11 |
| LSTM Zhang et al. (2019b) | | 13.07 | 13.19 | 14.13 | 13.69 | 12.84 | 12.35 | 12.15 | 12.99 |
| WReN Santoro et al. (2018) | | 17.94 | 15.38 | 29.81 | 32.94 | 11.06 | 10.96 | 11.06 | 14.54 |
| CNN Zhang et al. (2019b) | | 36.97 | 35.58 | 30.30 | 33.53 | 39.43 | 41.26 | 43.20 | 37.54 |
| ResNet-50 He et al. (2016) | | 53.43 | 52.82 | 41.86 | 44.29 | 58.77 | 60.16 | 63.19 | 53.12 |
| LEN Zheng et al. (2019) | | 72.90 | 80.20 | 57.50 | 62.10 | 73.50 | 81.20 | 84.40 | 71.50 |
| CoPINet Zhang et al. (2019b) | | 91.42 | 95.05 | 77.45 | 78.85 | 99.10 | 99.65 | 98.50 | 91.35 |
| **DCNet** | | **93.58** | **97.80** | **81.70** | **86.65** | **99.75** | **99.75** | **98.95** | **91.45** |
| Human | | 84.41 | 95.45 | 81.82 | 79.55 | 86.36 | 81.81 | 86.36 | 81.81 |

additional supervision, DCNet performs better than CoPINet and it achieves the best performance among all 7 figure configurations.

**Performance on PGM Dataset.** We employ the same experimental setup of CoPINet (Zhang et al., 2019b) that only reports the testing accuracy of the neutral regime of the PGM dataset, as it corresponds most closely to traditional supervised regimes. Due to the fact that the PGM dataset does not provide information about the figure configurations, we only report the average accuracy of each model. As the results reported in Table 1, the most popular network architectures fail to achieve good performance on the PGM dataset, including LSTM, CNN, and ResNet-50. In contrast, the abstract reasoning models, *i.e.* Wild-ResNet, WReN, LEN, and CoPINet, achieve better accuracy. Moreover, with auxiliary annotations about how the logical rules are applied, the performance of WReN and LEN is greatly improved, *i.e.* from 62.60% to 76.90% and 68.10% to 82.30%, respectively. However, such a learning strategy requires additional supervisory signals.

**Performance on Small Data Sizes.** In practice, supervised learning methods often require a large amount of well-labeled data for training, while humans are able to solve RPM problems without such massive training efforts. In order to evaluate the model generalization and learning efficiency of the models on abstract reasoning, we further report the testing accuracy of our DCNet when the

training set size shrinks on both RAVEN and PGM datasets. In this experiment, we use the same experimental setup of CoPINet (Zhang et al., 2019b) and randomly choose different subsets of the full training set and evaluate the average accuracy on the full test set. Besides, we compare with CoPINet to demonstrate the effectiveness of our method.[1]

Table 3: Model performance under different training set sizes on RAVEN dataset.

| Training Set Size | CoPINet | DCNet |
|---|---|---|
| 658 (1.57%) | 44.48 | **60.09** (+15.61) |
| 1, 316 (3.13%) | 57.69 | **71.91** (+14.22) |
| 2, 625 (6.25%) | 65.55 | **79.75** (+14.20) |
| 5, 250 (12.5%) | 74.53 | **84.42** (+9.89) |
| 10, 500 (25.0%) | 80.92 | **87.95** (+7.03) |
| 21, 000 (50.0%) | 86.43 | **91.31** (+4.88) |
| 42, 000 (100%) | 91.42 | **93.87** (+2.45) |

Table 4: Model performance under different training set sizes on PGM dataset.

| Training Set Size | CoPINet | DCNet |
|---|---|---|
| 293 (0.25%) | 14.73 | **15.94** (+1.21) |
| 1, 172 (0.10%) | 15.48 | **18.76** (+3.32) |
| 4, 688 (0.39%) | 18.39 | **25.78** (+7.39) |
| 18, 750 (1.56%) | 22.07 | **34.04** (+11.97) |
| 75, 000 (6.25%) | 32.39 | **43.10** (+10.71) |
| 300, 000 (25%) | 43.89 | **50.26** (+6.37) |
| 12,000,000 (100%) | 56.37 | **68.57** (+12.20) |

Table 3 shows the comparison results of CoPINet and DCNet on the RAVEN dataset. It can be seen that DCNet performs better than CoPINet on all subsets. Especially, when half the training samples are provided, DCNet achieves a comparable accuracy to CoPINet that trains on the full dataset, *i.e.* 91.31% *vs.* 91.42%. In addition, given 658 training samples, *i.e.* a 64× smaller dataset, DCNet achieves a testing accuracy of 60.09%, which outperforms CoPINet by a large margin of 15.61%.

Table 4 shows the comparison results of CoPINet and DCNet on the PGM dataset. Similar to the results observed in Table 3, DCNet also performs better than CoPINet on all subsets, up to 11.97% improvement when trained on 18, 750 samples. In addition, when DCNet is trained on a 64× smaller dataset (75K samples), DCNet outperforms LSTM, CNN and ResNet-50. Besides, DCNet achieves better accuracy than Wild-ResNet when trained on a 16× smaller dataset. Thus, the results shown in Table 3 and 4 demonstrate the good performance of DCNet when few training samples are provided.

**Generalization Test.** To measure the generalization ability of our method, we further conduct experiments on RAVEN to show the results on unseen figure configurations, see Table 5. Compared to DRT (Zhang et al., 2019a) that trains the model on one figure configuration and then tests it on similar cases, our experimental setup is more challenging, as we evaluate the model on all figure configurations, including very different ones. From Table 5, it can be seen that DCNet outperforms CoPINet on 5 figure configurations and is only a bit worse on the other 2 configurations, verifying the advantage of our method.

Table 6 shows the generalization test performance of our method on PGM. Following the experimental setup in MLRN (Jahrens & Martinetz, 2020), we report the performance on the interpolation and extrapolation regimes, details about these two regimes are introduced in (San-

Table 6: Generalization test on PGM.

| Method | neutral | interpolation | extrapolation |
|---|---|---|---|
| WReN | 62.6 | **64.4** | 17.2 |
| MLRN | **98.0** | 57.8 | 14.9 |
| DCNet | 68.6 | 59.7 | **17.8** |

toro et al., 2018). Although MLRN performs very well on the neutral regime, it is a slightly worse than WReN and DCNet on the interpolation. The worst generalization is observed on the extrapolation regime, where all compared methods fail to achieve good performance. Thus, robust abstract reasoning model with strong generalization ability still needs further investigation.

**Ablation Study.** To verify the effectiveness of each contrast module in our method, we conduct experiments on two datasets for ablation study, the results are reported in Table 1. Without the rule contrast module, the testing accuracy degrades 0.84% on RAVEN and 5.12% on PGM, respectively. With the rule contrast module, the proposed method is able to adapt to various logical rules of different RPM problems and further boost the performance, showing the effectiveness of our method.

---

[1]We only compare with CoPINet on small data sizes, the reasons being: (1) DCNet outperforms MXGNet with only 1/8 of the training samples on RAVEN dataset (*i.e.* 84.42% *vs.* 83.91%), see Table 1 and Table 3; (2) MLRN (Jahrens & Martinetz, 2020) did not report the accuracy on RAVEN and ACL (Kim et al., 2020) did not report the accuracy on PGM. Besides, ACL requires symbolic labels to increase the number of training samples for few-shot learning while we do not use such annotations.

Table 5: Generalization test on RAVEN. Each model is trained on row-wise configuration and tested on column-wise configurations.

| Method | Avg | Figure | Center | 2*2Grid | 3*3Grid | L-R | U-D | O-IC | O-IG |
|---|---|---|---|---|---|---|---|---|---|
| CoPINet | 66.71 | Center | 98.05 | 53.05 | 49.81 | 78.65 | 87.20 | 61.40 | 38.35 |
| | 68.19 | 2*2Grid | 64.55 | 71.20 | 67.65 | 87.25 | 83.00 | 62.25 | 41.45 |
| | 63.76 | 3*3Grid | 48.90 | 57.95 | 76.85 | 63.95 | 51.80 | 71.70 | 75.20 |
| | 69.83 | L-R | 55.50 | 51.90 | 54.55 | 98.25 | 92.35 | 76.40 | 59.85 |
| | 73.95 | U-D | 63.25 | 56.30 | 55.75 | 81.90 | 67.40 | 94.70 | 98.35 |
| | **68.81** | O-IC | 51.50 | 46.30 | 54.60 | 88.85 | 64.35 | 88.90 | 87.15 |
| | **68.74** | O-IG | 44.00 | 48.60 | 54.75 | 75.45 | 84.30 | 82.95 | 91.10 |
| DCNet | **70.76** | Center | 97.25 | 54.10 | 51.80 | 84.75 | 86.40 | 72.90 | 48.10 |
| | **74.09** | 2*2Grid | 69.35 | 76.60 | 66.55 | 86.85 | 88.60 | 72.45 | 58.20 |
| | **68.75** | 3*3Grid | 49.20 | 56.45 | 81.75 | 78.20 | 78.60 | 74.80 | 62.25 |
| | **73.21** | L-R | 60.30 | 53.35 | 59.30 | 98.50 | 91.70 | 83.85 | 65.45 |
| | **75.38** | U-D | 69.30 | 54.10 | 56.75 | 93.25 | 98.75 | 85.65 | 69.85 |
| | 68.11 | O-IC | 46.75 | 40.35 | 47.30 | 84.60 | 88.00 | 97.80 | 72.00 |
| | 66.36 | O-IG | 48.25 | 44.60 | 51.80 | 79.50 | 75.75 | 77.05 | 87.55 |

Without the choice contrast module, the testing accuracy of DCNet sharply degrades by 57.07% on RAVEN and 10.81% on PGM. The performance of DCNet on RAVEN degrades more than that of on PGM. The reason for such a difference is mainly the characteristic of dataset generation. As there is only one attribute change for the answer generation (Zhang et al., 2019a), the wrong choices in RAVEN are more similar to the correct answer when compared to the images in PGM. Thus, similar choices will degrade the correct answer prediction without the choice contrast.

## 4.5 DISCUSSION

We discuss some key questions on abstract reasoning as follows. (1) Following the recent abstract reasoning works (Santoro et al., 2018; Zhang et al., 2019a; Zheng et al., 2019; Zhang et al., 2019b; Wang et al., 2020), we also use neural networks to solve RPM. Since this is a black-box strategy, it does not provide certain rules to explain how the correct answer is predicted. Developing explainable models on RPM is an important direction in the future. (2) Because of similar problem structure (question and choices) in RPM and Visual Question Answering (VQA), we hope our work may help to improve the performance of some VQA problems and domain adaptation (Cao et al., 2019) on different visual reasoning tasks. (3) Unlike humans who are good at solving multiple abstract reasoning tasks, existing models mainly focus on a single task, *e.g.,* RPM questions in this work. For artificial general intelligence, benchmark datasets on multiple abstract reasoning tasks and appropriate evaluation metrics need further investigation.

## 5 CONCLUSION

In this work, we propose a Dual-Contrast Network (DCNet) towards solving RPM problems. Specifically, a rule contrast module is used to compare the latent rules among different rows/columns, and a choice contrast module is used to increase the relative differences of candidate choices. Without using auxiliary annotations and assumptions on hidden rules for model training, extensive experiments on the RAVEN and PGM datasets show that the proposed DCNet achieves the state-of-the-art accuracy. In addition, DCNet shows better performance when few training samples are provided. Experiments on generalization test show that robust abstract reasoning models with strong generalization ability needs further investigation.

### ACKNOWLEDGMENTS

This research is supported by the Agency for Science,Technology and Research (A*STAR) under its AME Programmatic Funding Scheme (#A18A2b0046). Tao Zhuo's research is also partially supported by National Natural Science Foundation of China (No. 62002188).

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

# A APPENDIX

## A.1 ROW AND COLUMN FEATURES

In this work, we consider both the row-wise and column-wise features, for each RPM problem $\boldsymbol{x}_i$, the formed rows $\boldsymbol{r}_i$ and columns $\boldsymbol{c}_i$ can be denoted as:

$$\boldsymbol{r}_{i,j} = \begin{cases} (\boldsymbol{x}_{i,1}, \boldsymbol{x}_{i,2}, \boldsymbol{x}_{i,3}), & j = 1 \\ (\boldsymbol{x}_{i,4}, \boldsymbol{x}_{i,5}, \boldsymbol{x}_{i,6}), & j = 2 \\ (\boldsymbol{x}_{i,7}, \boldsymbol{x}_{i,8}, \boldsymbol{x}_{i,j+6}), & j \geq 3, \end{cases} \qquad \boldsymbol{c}_{i,j} = \begin{cases} (\boldsymbol{x}_{i,1}, \boldsymbol{x}_{i,4}, \boldsymbol{x}_{i,7}), & j = 1 \\ (\boldsymbol{x}_{i,2}, \boldsymbol{x}_{i,5}, \boldsymbol{x}_{i,8}), & j = 2 \\ (\boldsymbol{x}_{i,3}, \boldsymbol{x}_{i,6}, \boldsymbol{x}_{i,j+6}), & j \geq 3, \end{cases} \qquad (5)$$

where $\boldsymbol{r}_{i,j}$ denotes the $j$-th row of $\boldsymbol{r}_i$, $\boldsymbol{c}_{i,j}$ denotes the $j$-th column of $\boldsymbol{c}_i$, $j \in \{1, \cdots, 10\}$, $\boldsymbol{x}_{i,j+6}$ represents $(j + 6)$-th image of $\boldsymbol{x}_i$.

## A.2 IMPLEMENTATION DETAILS

Similar to the experimental setup of previous works (Zhang et al., 2019b; Zheng et al., 2019; Wang et al., 2020), for general performance evaluation, we train the models on the training set, tune the parameters on the validation set, and report the accuracy on the testing set.

In the network architecture, the input dimension is $N * C * H * W$, where batch size $N = 32$, input channel $C = 3$ (as there are 3 gray-scale images in a row/column), image size $H = 96$ and $W = 96$. The output dimension of the feature encoder is $N * 128 * \frac{H}{4} * \frac{W}{4}$. Besides, the input feature dimension of MLP layer is 512.

## A.3 EXAMPLES AND RESULTS ON TWO DATASETS

Figure 3 shows the our testing accuracy of different training epochs, the proposed method achieves an accuracy of 83.08% on RAVEN with only one epoch, which has outperformed many state-of-the-art methods, *e.g.,* ResNet18+DRT (Zhang et al., 2019a) (59.56%) and LEN (Zheng et al., 2019) (72.90%), see Table 1. Compared to the most recent method MXGNet (Wang et al., 2020) (83.91% with more than 10 training epochs), our model is more efficient, since we exploit the inherent structures of the RPM problems with dual-contrast.

Figure 4 presents two examples of the RAVEN dataset (Zhang et al., 2019a) with two different figure configurations. Besides, Figure 5 shows two examples of the PGM (Santoro et al., 2018) dataset with different line and shape configurations. To demonstrate the robustness of the proposed method, the network architecture and all training parameters are kept the same on both two datasets.

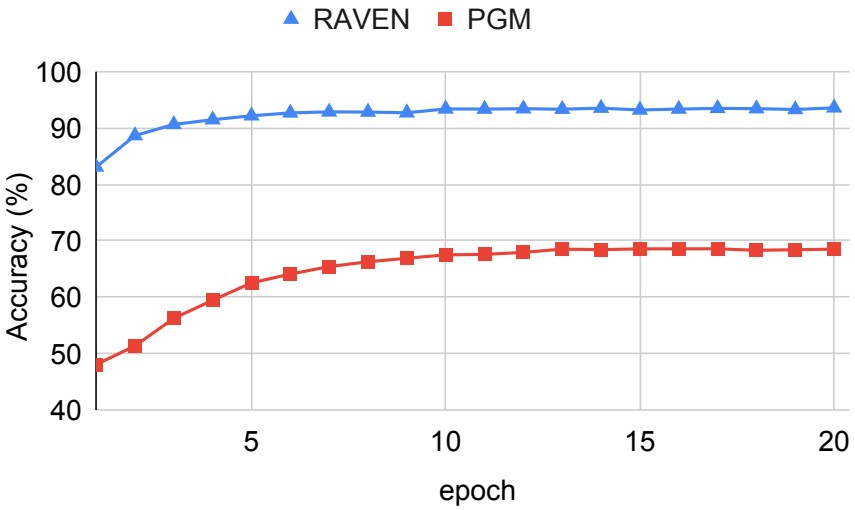

Figure 3: Testing accuracy of different training epochs on two datasets.

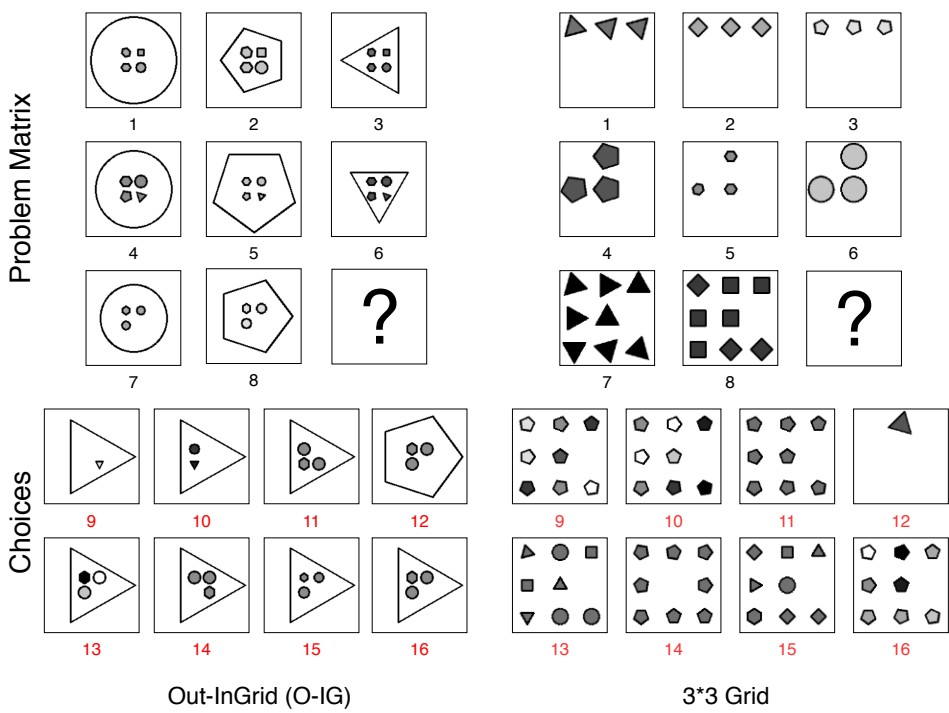

Figure 4: Two examples in RAVEN dataset (Zhang et al., 2019a). The correct answer is 16 (the left panel) and 11 (the right panel).

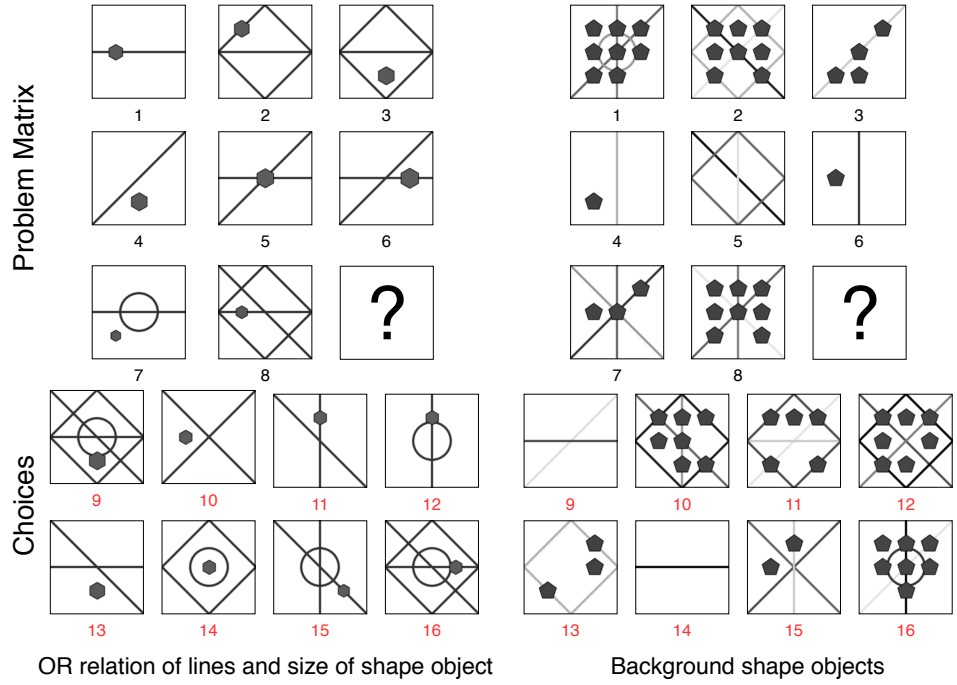

Figure 5: Two examples in PGM dataset (Santoro et al., 2018). The correct answer is 16 (the left panel) and 15 (the right panel).

