# OpenReview forum: "Effective Abstract Reasoning with Dual-Contrast Network"
_ICLR.cc/2021/Conference — ICLR 2021 Poster_

### Official Review · AnonReviewer4 · 2020-10-17
**Paper 576 Review**

**Rating:** 5
**Confidence:** 4

**Review:**

The paper proposed an inductive-bias contrast module that improves performance on Raven Progressive Matrices datasets. The proposed module makes use of the prior knowledge that rules can only exist in rows and columns of the diagram matrix.

While the paper claims to out perform previous SOTA, I have to point out that the authors are missing citations of the more recent SOTA methods on PGM and RAVEN datasets, such as [1] and [2]. If you take into account these more recent results, the proposed model only achieves slightly better results on RAVEN dataset, while not achieving SOTA results on PGM dataset.

There is some (but limited) novelty in using the prior knowledge of RPM rules to design inductive-bias module that works for this specific tasks. However the module is tailor designed for RPM tasks, which means it cannot be easily adapted to other types of reasoning tasks. Thus I feel the paper will only have impact within the RPM community, but little impact in the general ML and Reasoning community.

While the authors reported scores on PGM and RAVEN datasets, the authors did not report generalisation performances on various generalisation splits of PGM and RAVEN datasets. These missing results are in fact more interesting, as PGM and RAVEN datasets are designed for measuring generalisation capability of neural models.

In summary, I vote for reject as this paper is not citing the more recent SOTA results, is tailor-designed only for RPM tasks, and lacks experiments on the generalisation split of PGM and RAVEN datasets.

Clarity:
The paper is clearly written and easily understandable

-----------After Rebuttal--------------
I decide on up my score by 1 since indeed the authors are only required to cite papers that are peer-reviewed and published before 2nd August. I decide not to up my score any further because my other two concerns remain. I appreciate that the authors performed additional experiments. While the generalization performance is better than MLRN, it does not outperform other baselines such as MXGNet.


Reference:
1. Kim, Youngsung, et al. "Few-shot Visual Reasoning with Meta-analogical Contrastive Learning." NeurIPS 2020 (2020).
2. Wu, Yuhuai, et al. "The Scattering Compositional Learner: Discovering Objects, Attributes, Relationships in Analogical Reasoning." arXiv preprint arXiv:2007.04212 (2020).

---

> ### Comment · AnonReviewer2 · 2020-11-16
> **Prior work**
>
> My understanding is that we are not supposed to require comparison to prior work that was not published before the deadline. It seems like none of the papers that are the basis for this rejection rating were published before the deadline. Was the work you cited or any other important and missing prior work published before the deadline?

---

> > ### Comment · AnonReviewer4 · 2020-11-19
> > **About Prior work**
> >
> > Regarding the two previous work I mentioned, the first paper is published in this year's NeurIPS, while the second paper is not yet published. I think at least the first paper should be discussed as the NeurIPS acceptance announcement were before ICLR deadline. Both of these papers were uploaded to Arxiv in July and received some amount of attention.  I believe source codes are also publicly online.
> > My subjective judgment is that these paper should be discussed, but I am happy to discuss these if other reviewers and ACs think this is not appropriate.

---

> ### Author Response · Authors · 2020-11-20
> **Response to Reviewer #4**
>
> Thank you very much for the valuable comments.
> - The two papers mentioned by the reviewer are non peer-reviewed before the ICLR deadline. According to the ICLR review policy (https://iclr.cc/Conferences/2021/ReviewerGuide, 6th FAQ), we are not required to cite and compare with the very recent [1] (published on or after August 2) and non peer-reviewed papers [2]. The author notification of NeurIPS is September 25, but we do not know whether [1] is accepted before the ICLR deadline (October 2). Besides, both [1] and [2] use symbolic labels as auxiliary annotations for model training while we do not use such annotation.
>
> To demonstrate the effectiveness of our method, we have added [1] (denoted as ACL) for comparison. (1) As the results reported in Table 1, the accuracy of our method is only 0.16% less than ACL on RAVEN, but we do not use auxiliary annotations. Besides, ACL [1] did not report its performance on PGM dataset and its code is not released. (2) Technically, in order to increase the number of training samples for few-shot learning, ACL [1] uses symbolic labels to generate new RPM problems with the same problem semantics but with different attributes (e.g., shapes and colors). Compared to ACL, we do not use symbolic labels for model training and we do not increase training samples for few-shot learning.
> - We agree with you that this method is only applied for RPM tasks. We have discussed this issue in Section 4.5 and we currently follow the experiment setup of previous works (e.g. the baselines in Table 1). General applications on other visual reasoning tasks need further investigation.
> - Thank you very much for the good suggestions. As discussed in CoPINet, performance analysis on small data sizes can be also considered as generalization test, because many rules are not covered by a small number of training samples. To further show the effectiveness of our method, we conducted two experiments for the generalization test on RAVEN (Table 5) and PGM (Table 6) dataset. Table 5 shows the performance on unseen figure configurations, our method outperforms CoPINet on 5 cases and it is a bit worse on the other 2 cases. In Table 6, although MLRN is elaborately designed for the neutral regime of PGM performs very well (98.0%), our method achieves better performance than MLRN on the interpolation (59.7% vs. 57.8%) and extrapolation (17.8% vs. 14.9%) regimes of PGM.
>
> MLRN: Jahrens, M., & Martinetz, T. Solving Raven's Progressive Matrices with Multi-Layer Relation Networks. IJCNN, 2020.

---

### Official Review · AnonReviewer3 · 2020-10-28

**Rating:** 8
**Confidence:** 4

**Review:**

The paper proposes a new approach for abstract reasoning and explores it in the context of the RPM task. In contrast to other competing approaches, the authors seek to build into the model as few assumptions as possible to keep the model general and not specific to the specific problem or to particular annotations or supervision signals. The general capability that they seek to incorporate into the model is the ability to effectively compare and contrast candidates in tasks that require choosing the best fit.

The task is important, presented carefully and is well-motivated and the paper is clear and easy to follow. The related work section covers the necessary backgrounds including both visual reasoning and general and the RPM task in particular. It also presents the existing methods and discuss their disadvantages compared to the new approach - mainly their stronger reliance on supervision to learn good, semantic or disentangled features that will help most in addressing the task, or particular assumptions other approaches make about the specific properties or structure of the RPM task that may not hold in others or more general cases.

The task is also clearly presented and the authors explain how it consists of two main sub-tasks: (1) identifying the rule that links the existing rows or columns, and (2) comparing the candidate answers to choose the best fit. They propose network modules to solve each of these tasks correspondingly: a rule-contrast module, and a choice-contrast module, to complete each of these sub-tasks. For both  sub-stack they use clustering to find contrast and similarities between elements. This idea is simple, nice and quite novel I believe in these contexts.

Experiments are performed over two datasets of RAVEN and PGM, and achieve 5.77% improvement on average over state-of-the-art, and larger improvement in scarcer-data regimes, which the authors particularly focus on. They provide useful information about the datasets, baselines and implementation details and experiment settings, along with an ablation study to give further insight into the benefits of particular aspects of the model.

A particular comment that I have is that it seems that the model considers each answer by replicating the board k times for the k candidates, and considering each such completion alternative. While working for the RPM case, it won’t scale to problems where there’s a larger number of candidates, potentially even not-bounded. Restructuring the network to be able to preprocess as much as it can about the board so to reduce the amount of answer-dependent computation may be very useful to make the approach more general and efficient. I’m also looking forward to hearing about further applications of this idea as discussed at the end of the paper in the general VQA task or on other abstract reasoning, for instance such as the Abstraction and Reasoning Challenge.

Overall, Great work!

---

> ### Author Response · Authors · 2020-11-20
> **Response to Reviewer #3**
>
> Thank you very much for the positive comments and insightful suggestions.

---

### Official Review · AnonReviewer2 · 2020-11-01
**Interesting new approach to RPM**

**Rating:** 7
**Confidence:** 3

**Review:**


Summary
---
This paper proposes a new approach that improves performance on the Raven's
Progressive Matrices problem without auxiliary annotation by using two
complementary principles to structure inference.

(introduction)
The RPM problem is a visual reasoning problem where a 3x3 grid of images is provided
with one empty slot and 8 choices to fill in to that slot. The goal is to chose
the correct image via reference to the logical structure of the grid.
Two structures stand out:
1) The same rules apply to every row/column.
2) Many of the incorrect choices obey some, but not all of the rules.

(approach)
This paper incorporates both structures into its neural net design.
(structure 1) Each of the 2 provided rows and the 8 possible completions of the 3rd row
are embedded into a common space, and each choice is represented as a difference
between that choice's embedding and the embedding of the 2 known rows.
(structure 2) Next the approach considers the average embedding of the choices
and represents each choice by its position relative to that centroid.
This representation is fed through an MLP-sigmoid configuation to get a score
for each choice and perform the final classification.

(experiments)
Experiments compare to many recent state of the art baselines.
1. The proposed DCNet outperforms all baselines, without or without auxiliary supervision, on the RAVEN dataset.
2. The proposed DCNet outperforms all baselines, except those with auxiliary supervision, on the PGM dataset.
3. Performance degrades when the amount of training data is decreased, but not as much as CoPINet.
4. Ablations show that both of the structures incorporated into DCNet help improve performance.



Strengths
---

The RPM problem is interesting.

The dual contrast approach is well motivated.

The proposed DCNet outperforms relevant baselines on both datasets.

Ablations help understand the importance of the two types of structure and how performance improvements are distributed over subsets of the dataset.



Weaknesses
---

All the weaknesses I found in this paper were pretty minor. A few parts weren't as clear as I would have liked and the paper probably won't have much significance outside of the literature on RPM.


Clarity:

* The notation in the 2nd paragraph of 3.1 is a bit confusing. A figure or more specific example might help clarify. Is r_i,j a single image or all three images from a row concatenated together? Is each input image a different channel or are they concatenated?

* Multiple times the paper mentions that a disadvantage of the CoPINet approach is that it requires knowing the maximum number of possible rules and attributes. It would be nice if the paper elaborated on what kind of conern this is. Is this bad because it's not satisfying? (I agree that it isn't.) Or is this bad because of some practical issue? (Does it have a real effect on performance in the considered datasets?)

* Is one of the two contrasts generally enough to solve the problem in theory? Do these two contrasts just make inference easier, or do both really need to be considered by any system that solves the problem in general?


Significance:

* The correlation of RPM scores with general intelligence in humans makes the problem interesting on its own; more than a toy problem. But the approach is still very specific. I doubt very much from this approach will generalize to other problems.

* I understand what makes the approach different, but I don't have a qualitative assessment of how performance changed. What sorts of problems do other models fail at that this DCNet succeeds at? Maybe there's not an intuitive answer to that question which has been found in investigations so far. If so, I'd find it useful for the paper to say that.

* The performance improvements aren't very large, though I think they are probably significant, and that this approach would be a useful contribution even if it didn't beat all the baselines. Still, it would be useful to get a sense for the variance over random initializations and bootstrapping (i.e., confidence intervals).


Other:

* The choice of sigmoid instead of softmax in section 3.4 is a bit odd. Exactly one of the choices is correct, so it seems more natural to normalize them together instead of independently.


Preliminary Evaluation
---

Clarity - The paper is pretty clear overall, though some parts could be improved.
Quality - The approach is well motivated and the experiments include lots of ablations and comparisons to baselines that make the conclusion well supported.
Significance - This will likely have some small significance in the RPM literature. Improvements are small and the approach is fairly specific to RPM. However, the improvements are meaningul and the approach is interesting.
Originality - Previous approaches to RPM problems have taken advantage of the problem structure, but this specific structure in this way.

All of the factors above clearly point to acceptance, though general interest in this paper will be limited.



Suggestions
---

* The related work specifies that CoPINet also takes advantage of contrast. Since contrast is the central motivation of this paper, I wanted more detail about how exactly they do that. It would be nice to expand the related work to provide a bit more detail about contrast in CoPINet.

---

> ### Author Response · Authors · 2020-11-20
> **Response to Reviewer #2**
>
> Thank you very much for the insightful comments.
>
> Clarity:
> - $r_{i,j}$ is all three images from a row concatenated together and the input channel is 3. The details of our implementation have been described in Appendix A.2.
> - (1) Given an arbitrary RPM problems, the maximum number of possible rules and attributes should be unknown, as we do not know the details (e.g. how many colors, shapes in attributes) of the problem generation. (2) As CoPINet uses several linear layers to convert the features from attributes and rules to a new feature space with a fixed dimension (64), its performance is not sensitive to the assumption in the considered datasets.
> - In theory, the rule contrast may generally be enough, as it measures the differences of latent rules among all rows/columns (with filling choices). However, in practice, because of different strategies of problem generations and limited training samples, using the rule contrast alone may not perform well. Thus, we design another choice contrast to increase the relative feature differences of the third row/column (with filling choices). Based on these two contrasts, it can make the training and inference easier.
>
>
> Significance:
> - Thanks for the valuable comments. We agree with you that the current method is only applied on the RPM task. As discussed in Section 4.5, we currently follow the experiment setup of previous works (e.g. the baselines in Table 1). General applications on other visual reasoning tasks need further investigation.
> - It is indeed hard to intuitively point out what kinds of problems are absolutely solved by our method, because deep learning methods remain a black-box. Compared to other models, our method is motivated by the basic RPM problem formulation, thus our model can exploit the inherent structure of RPM and there is no special assumption about the latent rules.
> - Thank you for the good suggestions. We agree with you that the overall performance improvement of our method is not very large. However, as the results reported in Table 3 and 4, when a few training samples are provided, our method significantly outperforms CoPINet. For example, the accuracy of DCNet is comparable to CoPINet (91.31% vs. 91.42%) on RAVEN with only 50% training samples. Besides, DCNet slightly outperforms MXGNet (84.42% vs. 83.91%) with only 1/8 of the training samples. Therefore, following the evaluation metrics in previous methods, we did not report the confidence intervals.
>
> Other:
> - Thanks for pointing out this confusing description. The sigmoid normalization of our method is also processed in the natural way. We have clarified this issue in Section 3.4.
>
> Suggestion:
> - More discussions about the contrast used in CoPINet are added in the revision, see Section 2.2. Different from the "objective-level contrast'' in Zhang et al. (2019b) that treats the complete problem matrix (9 images, including a filling choice) as an object for comparison, our model consists of a rule contrast module and a choice contrast module and we only compare the features of different rows/columns (3 images). Thus our method is more effective in exploiting the inherent structure of RPM on rows/columns.

---

### Official Review · AnonReviewer1 · 2020-11-02
**Interesting approach, impressive results**

**Rating:** 7
**Confidence:** 3

**Review:**

Summary:

The paper proposes a neural network based approach called Dual-Contrast Network (DCNet) to solve Raven’s Progressive Matrices (RPM). The approach consists of a rule contrast module that compares the latent rules between the unfilled (third) row/column and the filled (first and second) rows/columns, a choice contrast module that helps in picking the correct choice among the given eight choices, and finally uses a 2-layer MLP to predict scores for the choices. Different from previous approaches, the only supervision used in the proposed approach is the ground-truth choice. The approach achieves state-of-the-art performance on RAVEN and PGM datasets.

——————————————————————————————————————————————————————————————


Strengths:

S1) The approach achieves state-of-the-art performance.

S2) The approach requires the least amount of supervision — the ground-truth option, unlike previous works that require auxiliary annotations or assumptions.

S3) The paper shows the contribution of the approach’s two main modules using ablation studies.

S4) The approach outperforms previous works when trained with smaller training datasets.

——————————————————————————————————————————————————————————————


Weaknesses:

W1) The results lack standard deviation or error bars in the tables. Are the differences from previous works statistically significant?

W2) I think the paper would benefit a lot with further analysis into the results — what type of mistakes do previous methods tend to make which the proposed approach is able to overcome? The paper should contain qualitative examples showing what options previous methods predict, especially when they fail and the option predicted by the proposed approach.

W3) Similarly, the paper should analyze the failure modes of the proposed approach. Is the task solved? If not, what obstacles still remain?

W4) How does the computation complexity of the proposed approach compare with previous works?

W5) The paper claims that the ability to solve RPM correlates well with human intelligence. But the few examples shown in the paper seem super hard. How well do humans perform on these datasets? I believe it should be much lower than that of the previous works as well as the proposed approach. If that’s the case, why do we want to build better and better approaches for this task?


——————————————————————————————————————————————————————————————
——————————————————————————————————————————————————————————————

Update after rebuttal: I thank the authors for their responses to my questions. They satisfactorily answer most of my concerns. Overall, I agree with the concern that the proposed approach is specific to RPM and it's unclear how well it (or parts of it) would generalize to other problems but I think the approach is quite interesting, novel and achieves state-of-the-art results. Hence, I think the contributions of the paper are significant for acceptance.

---

> ### Author Response · Authors · 2020-11-20
> **Response to Reviewer #1**
>
> Thank you very much for the valuable comments.
> - W1: Currently, we follow the previous works (e.g. baselines in Table 1) for performance evaluation and we only report the average testing accuracy. Compared to standard deviation or error bars, performance on few-shot learning is more important (see Table 3 and 4) on RPM task, as humans do not require such massive training efforts. Besides, we added the generalization tests on two datasets for comparison, see Table 5 and 6.
> - W2: Thanks for the good suggestions. It is hard to say what kinds of problems are definitely solved by our method. Therefore, even if we give some qualitative examples for comparison, it does not mean that certain problems are completely solved. Therefore, we statistically analyze the performance from different perspectives, e.g. training the model on small sizes.
> - W3 and W5: Thank you very much for the insightful comments. The human performance is 84.41% obtained from Zhang  et al. (2019a)) on RAVEN (updated in Table 2) and it is not reported on PGM. Although the average accuracy of recent approaches has outperformed the human-level performance, it does not necessarily mean that this task is solved. (1) As discussed in Section 4.4, supervised learning methods often require a large amount of well-labeled data for training, while humans are able to solve RPM without such massive training efforts (sometimes with no training). When a few training samples are available, the performance of supervised approaches is still worse than humans, see Table 3 and 4. (2) The generalization ability of current methods is still poor, see Table 6. (3) Humans can explain how the answer is obtained with certain rules (e.g. shape, color, size, position, AND, XOR), but the recent approaches are black-boxes (Section 4.5), which cannot explain the output with hidden rules, see Section 4.5. (4) The long-term purpose of solving RPM is to study the visual reasoning capability of machines. As mentioned in previous works (e.g. WReN and DRT), studying RPM can help to further understand and improve the contemporary computer vision systems, including the structural, relational and analogical reasoning. Besides, since the logical rules of RPM can be understood by humans, it can also help to improve the interpretability of black-box models.
> - W4: The computation complexity of our method is comparable to that of CoPINet, which also consists of several ResBlock and CNN layers.

---

### Public Comment · ~Taylor_Whittington_Webb1 · 2020-11-11
**Significant previous results overlooked**

This work overlooks some previous results that significantly undermine its key claims. Specifically:
* In addition to the work cited by reviewer 4, Jahrens and Martinetz [1] showed that multi-layer relation networks can achieve an overall test accuracy of 98.03% on the PGM dataset, which is quite a bit higher than the results presented here (68.57%). Notably, those results were also achieved without including any of the auxiliary information emphasized in this paper.
* The other dataset employed in this work, RAVEN, has been shown by Hu et al. [2] to contain a bias that can dramatically inflate the performance of certain models. Specifically, the method for generating the answer set makes it such that a model can achieve very high accuracy without even observing the problem (e.g. Hu et al. showed that a ResNet could achieve 90.1% accuracy on this dataset while only observing the answer sets). For this reason, more recent work has used the 'Balanced-RAVEN' dataset introduced by Hu et al., which is much more difficult and cannot be trivially solved by observing only the answer set. Wu et al. [3] achieve better test accuracy (95.0%) on this more challenging version of the dataset than the model in this paper achieves on the original biased version (93.58%). Furthermore, it seems likely that the model presented in this work, which computes the difference for an embedding of each answer choice from the average of the embeddings for all choices, achieves high test accuracy on the original version of the RAVEN dataset by exploiting this bias, rather than reasoning about the problems themselves. This model needs to be tested on the Balanced-RAVEN dataset to determine whether that's the case.

1. Jahrens, M., & Martinetz, T. (2020). Solving Raven's Progressive Matrices with Multi-Layer Relation Networks. arXiv preprint arXiv:2003.11608.
2. Hu, S., Ma, Y., Liu, X., Wei, Y., & Bai, S. (2020). Hierarchical Rule Induction Network for Abstract Visual Reasoning. arXiv preprint arXiv:2002.06838.
3. Wu, Y., Dong, H., Grosse, R., & Ba, J. (2020). The Scattering Compositional Learner: Discovering Objects, Attributes, Relationships in Analogical Reasoning. arXiv preprint arXiv:2007.04212.

---

> ### Comment · AnonReviewer2 · 2020-11-16
> **Previous results**
>
> Have any of these been published? How long were they published before the ICLR 2021 deadline? My understanding is that we are not supposed to require comparison to prior work that was not published before the deadline.

---

> > ### Public Comment · ~Taylor_Whittington_Webb1 · 2020-11-17
> > **previous results**
> >
> > [1] is published at IJCNN 2020. It appears that [2] and [3] are only available as preprints.

---

> ### Author Response · Authors · 2020-11-20
> **Response to Taylor Whittington Webb**
>
> Thank you very much for pointing out these works.
> - As MLRN [1] is a very recent work (conference date of IJCNN 2020 is Jul 19, 2020 - Jul 24, 2020), which is missing in our previous version. We have added this method in our revision for comparison, see Table 1 and 6. (1) As discussed in Section 4.4 (page 6), MLRN [1] is elaborately designed for the neutral regime of PGM and it achieves an accuracy of 98.03%. However, the testing accuracy of MLRN is worse than the proposed DCNet on the interpolation (57.8% vs. 59.7%) and extrapolation (14.9% vs. 17.8%) regimes of PGM, see the generalization test results in Table 6.
> MLRN did not report the testing accuracy on RAVEN dataset. For further comparison, we run the released code of MLRN on RAVEN dataset with its default parameters. As the results reported in Table 1, MLRN performs bad (12.50%) on RAVEN. In contrast, DCNet achieves good accuracy (93.58%) and all the network architecture and training parameters are kept the same on both two datasets (see Section 4.1).
> - [2] and [3] are non peer-reviewed papers. According to the ICLR review policy (https://iclr.cc/Conferences/2021/ReviewerGuide, 6th FAQ), we are not required to cite and compare with these two papers. Besides, notice that the 'Balanced-RAVEN' [2] is not a regular benchmark dataset for performance evaluation. For example, the most recent paper (Kim, Youngsung, et al. "Few-shot Visual Reasoning with Meta-analogical Contrastive Learning." NeurIPS 2020) did not use it for evaluation.

---

### Decision · Program_Chairs · 2021-01-07
**Final Decision**

**Decision:**

Accept (Poster)

**Comment:**

This work proposes a new architecture for solving Ravens Progressive Matrices (RPM), a well known form of visual reasoning problem. The method relies on an operation that directly compares the final row and column (completed with different candidate answers) with the first two rows and columns. Doing this allows the network to perform better than previous approaches when measured on an in-distribution test set on two RPM datasets, RAVEN and PGM.

As the reviewers pointed out, a strength of this work is the strong performance on the neutral split of these datasets and the fact that the methods do not (unlike some other approaches) require access to any annotations from the dataset other that knowledge of the structure of the RPM task and access to the candidate answers and the correct answer.

However, a noted weakness is the fact that the network reflects the structure of the task more directly than other approaches, which means that the insight is specific to the problem of solving RPMs. Another weakness is that the authors focus on the neutral (in-distribution) splits. Reading the PGM paper, it is clear that the neutral split is not really the main focus of that dataset (it accounts for only 1/7 of the dataset), which seems to have been specifically developed as a benchmark for out-of-distribution generalisation. Indeed, the whole point of the RPM task is to measure the ability to induce abstract rules and principles from pixels, but without measuring out-of-distribution generalisation, can we really claim that any model has induced a 'rule'?

The authors mitigate this issue to a small degree during the rebuttal by adding scores on the 'interpolation' and 'extrapolation' splits of the PGM dataset, but still do not consider the other splits where rule application is most clearly tested.

I note that the weakness described above also applies to lots of other published work involving PGM and RAVEN datasets.

In summary, this is a well-executed, neat piece of work that shows a better way to fit a large dataset by incorporating knowledge of the structure of the data into the task. Because it does not consider the full benchmarks, only the in-distribution splits, it falls short of showing that this enables better induction of abstract principles or rules. On the majority opinion of the reviewers, and because there are no scientific flaws in the work, I recommend acceptance with weak confidence pending wider calibration across the program.